# Epidemiology of soil-transmitted helminths following sustained implementation of routine preventive chemotherapy: Demographics and baseline results of a cluster randomised trial in southern Malawi

Stefan Witek-McManus[1]*, James Simwanza[2], Alvin B. Chisambi[2], Stella Kepha[1,3,4], Zachariah Kamwendo[2], Alfred Mbwinja[5], Lyson Samikwa[5], William E. Oswald[1], David S. Kennedy[1], Joseph W. S. Timothy[1], Hugo Legge[1], Sean R. Galagan[6], Mira Emmanuel-Fabula[6], Fabian Schaer[7], Kristjana Ásbjörnsdóttir[8], Katherine E. Halliday[1], Judd L. Walson[6,8,9], Lazarus Juziwelo[10], Robin L. Bailey[11], Khumbo Kalua[2,5‡], Rachel L. Pullan[1‡]

1 Department of Disease Control, London School of Hygiene & Tropical Medicine, London, United Kingdom,
2 Blantyre Institute for Community Outreach, Lions Sight First Eye Hospital, Blantyre, Malawi, 3 Eastern and Southern Africa Centre of International Parasite Control, Kenya Medical Research Institute, Nairobi, Kenya,
4 Pwani University Bioscience Research Centre, Pwani University, Kilifi, Kenya, 5 College of Medicine, University of Malawi, Blantyre, Malawi, 6 Department of Global Health, University of Washington, Seattle, Washington, United States of America, 7 DeWorm3, Division of Life Sciences, Natural History Museum, London, United Kingdom, 8 Department of Epidemiology, University of Washington, Seattle, Washington, United States of America, 9 Department of Medicine and Department of Paediatrics, University of Washington, Seattle, Washington, United States of America, 10 National Schistosomiasis and STH Control Programme, Community Health Sciences Unit, Ministry of Health & Population, Lilongwe, Malawi, 11 Department of Clinical Research, London School of Hygiene & Tropical Medicine, London, United Kingdom

‡ These authors are joint senior authors on this work.
* stefan.witek-mcmanus@lshtm.ac.uk

## Abstract

Malawi has successfully leveraged multiple delivery platforms to scale-up and sustain the implementation of preventive chemotherapy (PCT) for the control of morbidity caused by soil-transmitted helminths (STH). Sentinel monitoring demonstrates this strategy has been successful in reducing STH infection in school-age children, although our understanding of the contemporary epidemiological profile of STH across the broader community remains limited. As part of a multi-site trial evaluating the feasibility of interrupting STH transmission across three countries, this study aimed to describe the baseline demographics and the prevalence, intensity and associated risk factors of STH infection in Mangochi district, southern Malawi. Between October-December 2017, a community census was conducted across the catchment area of seven primary healthcare facilities, enumerating 131,074 individuals across 124 villages. A cross-sectional parasitological survey was then conducted between March-May 2018 in the censused area as a baseline for a cluster randomised trial. An age-stratified random sample of 6,102 individuals were assessed for helminthiasis by Kato-Katz and completed a detailed risk-factor questionnaire. The age-cluster weighted prevalence of

**Data Availability Statement:** Data cannot be shared publicly at the time of publication because the study remains blinded to outcome data. Data will be made available on request through the LSHTM Data Compass (contact via the Site Administrator, researchdatamanagement@lshtm. ac.uk) for researchers who meet the criteria for access to these data. Requests for release of the data will be reviewed by the relevant institutional review boards.

**Funding:** The DeWorm3 Project is funded by a grant from the Bill & Melinda Gates Foundation (OPP1129535) TL, JLW. SK is supported by THRiVE- 2, a DELTAS Africa grant #DEL-15-011 from Wellcome Trust grant #107742/Z/15/Z and the UK Government. The funders had no role in study design, data collection and analysis, decision to publish, or preparation of the manuscript.

**Competing interests:** The authors have declared that no competing interests exist.

any STH infection was 7.8% (95% C.I. 7.0%-8.6%) comprised predominantly of hookworm species and of entirely low-intensity infections. The presence and intensity of infection was significantly higher in men and in adults. Infection was negatively associated with risk factors that included increasing levels of relative household wealth, higher education levels of any adult household member, current school attendance, or recent deworming. In this setting of relatively high coverage of sanitation facilities, there was no association between hookworm and reported access to sanitation, handwashing facilities, or water facilities. These results describe a setting that has reduced the prevalence of STH to a very low level, and confirms many previously recognised risk-factors for infection. Expanding the delivery of anthelmintics to groups where STH infection persist could enable Malawi to move past the objective of elimination of morbidity, and towards the elimination of STH.

 **Trial registration:** NCT03014167.

## Author summary

The major public health strategy to control soil-transmitted helminths (STH) is preventive chemotherapy, whereby those at greatest risk of morbidity–including children and women of childbearing age–are presumptively treated with a safe, effective and inexpensive anthelminthic drug. In Malawi, this has been successfully sustained for nearly a decade through annual school-based deworming, in addition to integration within child health campaigns and antenatal care. Routine surveillance of schoolchildren demonstrates that STH has been reduced to very low levels in this age group, but few community-based epidemiological surveys have been conducted to investigate STH in the broader population. In this study, we observed that while infection with STH has been reduced to low levels overall, it is much higher in adults and particularly in males, with the odds of being infected greater in those from less wealthy households or from households with lower levels of adult education. These results underline that while preventive chemotherapy has likely been key to reductions in STH; sub-populations not routinely targeted by preventive chemotherapy, and the most disadvantaged members of society, continue to be disproportionately affected. We propose that evaluation of more comprehensive control strategies–such as entire-community deworming–could overcome these limitations, and present a route to STH elimination.

## Introduction

Over the past two decades, the predominant approach to the control of soil-transmitted helminths (STH) has been the periodic administration of anthelmintic medicines to populations considered at greatest risk of morbidity–pre-school and school-age children, adolescent girls, and women of reproductive age–using an approach referred to as preventive chemotherapy (PCT). Aligned with global recommendations by the World Health Organization (WHO), control programmes coordinated by Ministries of Health and Education in partnership with non-governmental organizations (NGOs) and multi-lateral agencies have rapidly increased treatment coverage of pre-school aged (PSAC) and school aged children (SAC) from less than 15% of the global at-risk population in 2005 to nearly 70% by 2017 [1]. It has been estimated that this scale up has averted the loss of more than 500,000 disability-adjusted life years through sustained reduction in the intensity of STH infections [2].

In many STH endemic countries, including Malawi, control programmes routinely leverage up to four established mechanisms to reach these target populations with anthelmintics: at primary schools, during child-health campaigns, during mass drug administration for lymphatic filariasis, and within maternal health services. Since 2004, the National Schistosomiasis Control Programme (NSCP) of Malawi has included albendazole when implementing annual school-based delivery of praziquantel for schistosomiasis, with community-based mop-up days targeting non-attending SAC. In tandem, biannual "Child Health Days" deliver a package of health services to children under five years old that includes albendazole [3]. Treatment coverage for STH has been consistently high in both groups, with national coverage of 92% of PSAC and 75% of SAC reported to have received treatment in 2017 [4]. There has been substantial scale-up of focused antenatal care, which routinely includes anthelmintic treatment after the first trimester [5]. While implementation of routine antenatal care (ANC) has been erratic [6], the 2015–16 Demographic and Health Survey (DHS) reported that 52% of women took deworming medication during their last pregnancy [7]. Additional small-scale deworming continues to be carried out ad-hoc at sub-national levels, but may not be consistently reported [8,9]. Historic programmes, including the National Programme to Eliminate Lymphatic Filariasis (NPELF) have previously delivered albendazole (with ivermectin) annually to all individuals over the age of 5 years between 2008–2013 [10].

Together, these approaches can be viewed as a comprehensive STH control strategy that has resulted in the sustained delivery of anthelmintic treatment at consistently high levels to much of the population of Malawi for more than a decade. Accordingly, published literature on the prevalence of STH infection within Malawi broadly describes a setting where STH infection has declined during the past two decades, with the most recent national community-based survey conducted in 2011 reporting an STH prevalence of 0.3–3.8% [11]. It is therefore likely that the recently announced WHO 2030 targets for STH control programmes of "achieving and maintaining elimination of STH morbidity in PSAC and SAC" has been reached in Malawi [12]. To sustain and build upon these gains, it is crucial that routine sentinel monitoring surveys conducted with school-going children are accompanied by periodic comprehensive age and sex stratified epidemiological surveys. Such surveys can highlight demographic groups where STH infection continue to persist, identify contemporary and contextually relevant risk-factors, and can support informed and rational evaluation of the current STH control strategy, as demonstrated in similar settings elsewhere [13,14].

Here, we describe the profile of STH infection across communities in Mangochi district, southern Malawi, a previously highly endemic area, and investigate environmental, household and individual factors in order to identify population groups that remain at highest risk of infection. The data presented comprise a community census and cross-sectional parasitological survey conducted in 2018 as a baseline for the Deworm3 Malawi trial, an ongoing evaluation of the feasibility of interrupting STH transmission through biannual community-based MDA [15].

## Methods

Reporting of this study has been verified in accordance with the Strengthening the Reporting of Observational Studies in Epidemiology (STROBE) checklist [16].

### Ethical considerations

The parent trial of this study is registered at *ClinicalTrials.gov* (NCT03014167). This study was approved by the College of Medicine Research Ethics Committee (COMREC) at the University of Malawi (P.04/17/2161), the London School of Hygiene & Tropical Medicine (LSHTM)

Observational/Interventions Research Ethics Committee (12013) and the Human Subjects Division at the University of Washington (STUDY00000180). Prior to beginning the study, community engagement activities were conducted with respect to traditional leadership structures. Senior study officers conducted planning meetings with the Area Development Committees (ADC) responsible for the study site, who subsequently met with the respective Group Village Headman (GVH) and Village Headmen to coordinate engagement activities at the village level.

At the time of the community census, written informed consent was sought from the head of household or other adult household member before answering the questionnaire; for the parasitological survey, consent was sought from the individual selected to provide the stool sample and complete the individual-level questionnaire. Parental consent was sought for participants between 2 and 15 years and written assent was additionally obtained from participants aged 7 to 15 years. All information and consent procedures were conducted in relevant local languages (Chichewa or Chiyao). Where a participant was willing to give consent but was not functionally literate in the relevant language to complete the written consent form, a literate and impartial (i.e. not directly associated to any study staff member) witness of the participants choice was invited to witness the consent process. Where no impartial literate witness could be identified, the village volunteer accompanying the fieldworker conducting the informed consent process was eligible to act as witness if they were functionally literate; and where both neither a functionally literate impartial witness or village volunteer was available, any impartial witness of the participants choice fulfilled this role.

Treatment of participants identified as positive for STH was administered by the relevant community health worker, known as a *Health Surveillance Assistant* (HSA) using albendazole (400mg). Participants residing in a cluster subsequently randomised to the intervention arm of the trial were expected to receive this treatment during the first round of the trial intervention alongside all other eligible community members, and participants residing in a cluster that was randomised to the control arm of the trial were individually visited to be treated if (i) aged 2–14 years identified with an STH infection of any intensity or (ii) aged ≥15 years and identified with an STH infection of moderate or high intensity (MHI). All other infections identified (e.g. *Schistosoma* spp.) were treated as per national guidelines.

### Study setting and population

This study was conducted within Namwera zone, one of five health services provision units located in Mangochi district in the southern region of Malawi (Fig 1). Mangochi is a relatively rural area of Malawi, with 55% of the population aged under 18 years [17]. In line with regional and national trends, household use of basic sanitation and protected water sources is high, exceeding 90%. In contrast to other districts of Malawi, the majority ethnic group of the district is Yao and most common religion is Islam. The predominant source of livelihood is *ganyu* or informal off-farm labour, and the district has the lowest literacy rate nationally [17]. Following certification by WHO of the elimination of lymphatic filariasis (LF) in 2015 through community-based mass administration of albendazole and ivermectin, the district remains endemic for STH and schistosomiasis.

### Community census design

A community census was conducted between October-December 2017 by the study team in collaboration with the Ministry of Health & Population (MoH&P). A total of 124 villages were censused in Namwera, based on pre-established community health worker implementation units ('HSA catchment areas') linked to seven primary healthcare facilities that serve the area.

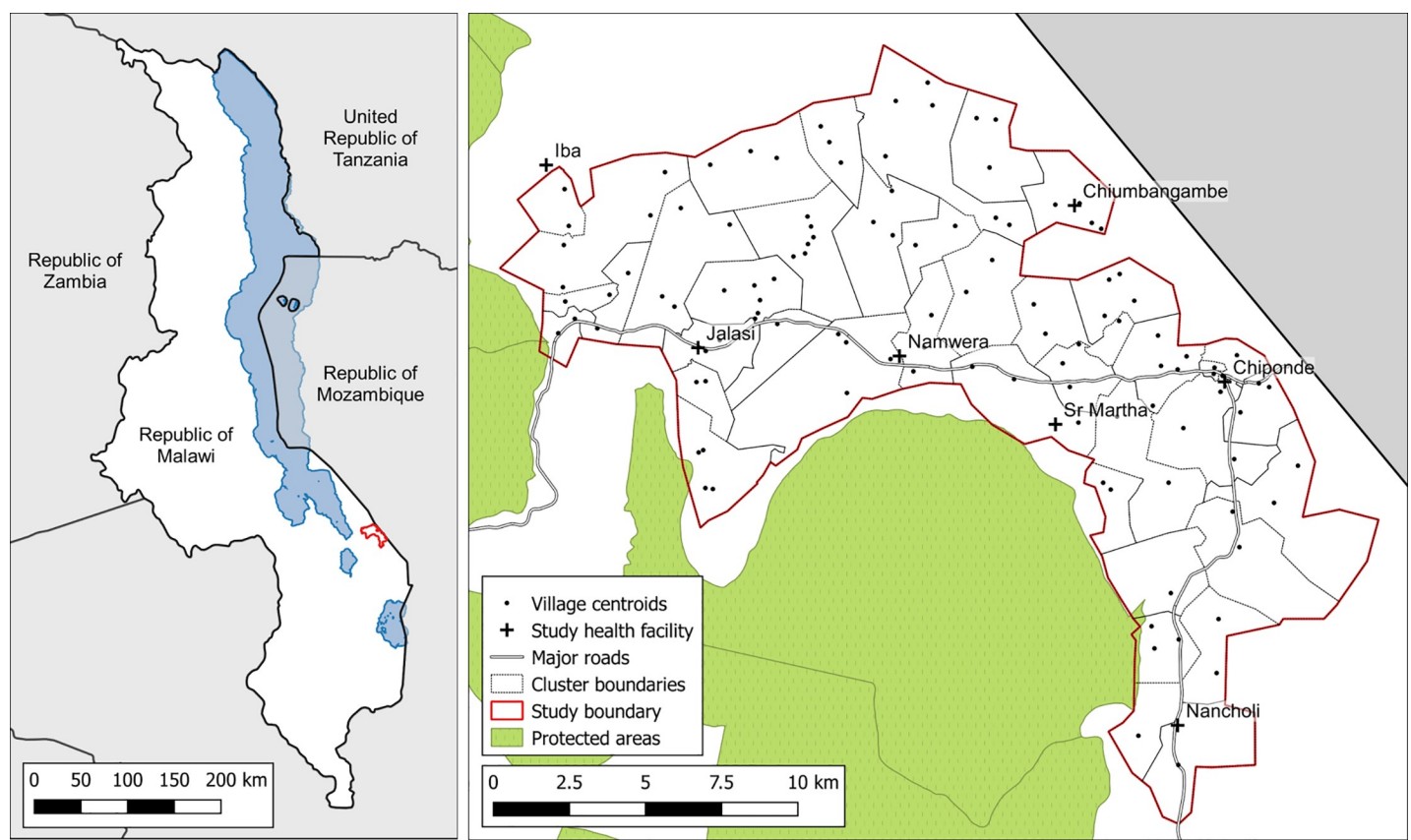

**Fig 1.** Location of study site within Malawi (left panel) and demarcated study clusters (right panel). Contains information from OpenStreetMap and OpenStreetMap Foundation, which is made available under the Open Database License.

In each village, a trained enumerator accompanied by a village volunteer administered a standard household questionnaire to an adult member of each household, following provision of informed written consent. If at the time of the initial visit either all adult members of the household were absent, or no adult member was available to consent or respond to the questionnaire, at least two further re-visits were conducted to complete the questionnaire. For all inhabited and consenting households, each member was individually enumerated and their residential status and school enrolment or education level recorded. Household-level information was collected using both reported (e.g. asset and livestock ownership, source of livelihood, and access to water and sanitation services) and observed (e.g. materials used for household construction) measures. Censused household were provided with a study ID card to facilitate household identification and linkage in subsequent study activities. Census questionnaire responses were recorded electronically using the SurveyCTO platform (Dobility, Inc) and GPS coordinates of all censused households, in addition to vacant and non-residential structures, were collected. Quality control of the census was conducted through random spot check by fieldwork supervisors and re-visits of 10% of censused households conducted by senior enumerators.

## Cluster delineation

Prior to demarcation, two censused villages were excluded due to their geographical isolation from the study site. Using the household GPS coordinates collected during the census, the

remaining villages were allocated into 40 study clusters, with priority given to preserving HSA catchment areas. Where an HSA catchment met the required cluster population size (1,650–4000 individuals) and was comprised of contiguous villages, the implementation unit served as the cluster. Where an HSA catchment was smaller than the required cluster population size, or was comprised of villages that were not contiguous, villages served by different community health workers were combined to form a cluster within the required population size. Where an implementation unit was larger than the required cluster population size, single villages were classified as a cluster. No villages were sub-divided during this process.

## Parasitological survey design

A parasitological survey of the study site was conducted between March and June 2018. The sample size determination has previously been described, and resulted in the selection of 150 individuals in each of 40 clusters [15]. In each cluster, an age-stratified sample of individuals (30 pre-SAC (1–4 years of age), 30 SAC (5–14 years of age) and 90 adults (≥15 years of age) were randomly selected using the community census as a sampling frame. Each sampled individual was approached at their household by a trained enumerator accompanied by a village volunteer and invited to participate in the parasitological survey. To maximise opportunity for recruitment, individuals were re-visited at their household at least two further times if they were absent or unavailable at the initial visit. Sampled individuals who could be located were invited to participate in the parasitological survey following confirmation that they were eligible (aged over 12 months, still resident within the study site, and did not plan to migrate outside the study site within five years). At the household level, reported water storage was recorded and structured observations of water storage and handwashing facilities conducted. An individual-level questionnaire with each enrolled participant included reported deworming treatment in the past year, observed shoe wearing, and latrine usage. Where sampled individuals reported using a toilet facility, observations were made on toilet facilities. Survey questionnaire responses were recorded electronically using the SurveyCTO platform. Following completion of the questionnaire, participants were requested to provide a single stool specimen. Specimens were collected the same day or early the following morning, with two further follow-up visits conducted where a specimen was not provided or was not suitable. Once >80% of sampled individuals had been recorded as not located, no longer resident, not eligible or refused to consent, sequential lists of replacement individuals (15 pre-SAC, 15 SAC and 45 adults) were provided to enumerators until the target sample size in each age category was achieved.

## Parasitological assessments

Stool specimens were placed into cooler boxes and transported to a field laboratory within eight hours of being collected from the participant. Specimens were then prepared for examination by Kato-Katz thick-smear method [18]. Two slides were prepared per specimen and read in duplicate by pairs of independent technicians between 30 to 60 minutes after slide preparation. A second slide reading for *S. mansoni* was conducted between 18 and 24 hours following the first reading per recommended practice [19]. Egg counts for each STH species, Schistosoma *spp*. and other helminth species were recorded separately. Infection was defined as the presence of at least one egg on at least one slide, confirmed by at least two laboratory technicians. Intensity was expressed as the arithmetic mean of eggs per gram (epg) of faeces across the two slides, categorised according to WHO classifications [20]. A random sample of 10% of all readings were re-read by a senior technician for the purpose of quality control. Following microscopic examination, a set of three aliquots containing 500mg of whole stool

sample suspended in 1ml of 95% ethanol were prepared in a separate area of the laboratory and cryopreserved at -80°c for future parasite DNA extraction and qPCR analysis.

### Environmental covariates

Analysis considered a suite of environmental and topographic conditions previously identified as potential drivers of STH transmission [13]. Data sources included: Enhanced Vegetation Index (EVI) and Land Surface Temperature (LST), produced by processing satellite images provided by the Moderate Resolution Imaging Spectroradiometer (MODIS) instrument operating in the Terra spacecraft (NASA) at a resolution of 250m; elevation and aridity at $1km^2$ from the Consortium for Spatial Information (CGIAR-CSI); soil acidity (pH KCl) and sand content from soilgrids.org at a resolution of 250m [21]. Environmental, topographic and population measures were extracted using point-based extraction for each household using Arc-GIS 10.3 (Environmental Systems Research Institute Inc. Redlands, CA, US). Estimates of population density per square kilometre were constructed by summing the total number of individuals within a $1km^2$ buffer around each household in ArcGIS. For households near study area boundaries, areas of the buffer that fell outside of the study area were removed and the population density was calculated by the number of individuals within the buffer divided by the remaining area of the buffer.

### Analysis

Data management and analyses were performed using Stata 16.1 (StataCorp, 2019; College Station, TX, USA). Information on ownership of household assets was used to construct a wealth index for each household using principal component analysis (PCA). Variables used to construct the final PCA included materials used to construct the household dwelling roof (grass thatch, metal), walls (fired brick, covered unfired brick) windows (none) and door (wooden planks); and household ownership of a cooking stove, mobile phone, mattress, bed and bicycle (average inter-item correlation= 0.27, alpha= 0.80). The first principal component accounted for 35.8% of the total variance. The indices were divided into quintiles within each setting. Household factors potentially associated with infection outcomes, including toilet facilities and household flooring, were not included in the wealth index to allow for independent assessment. Classification of household water, sanitation and handwashing facilities (WASH) was done according to WHO/UNICEF Joint Monitoring Programme (JMP) guidelines [22,23].

Age- and cluster-population weighted estimates were calculated using the proportion of the censused population living in the cluster. Prevalence and intensity descriptive analyses used robust standard errors to account for clustering within villages. Risk factor analyses were conducted for hookworm; *T. trichiura* and *A. lumbricoides* infection were not included due to very low infection levels found in the survey population. Univariable associations between presence of infection and individual-, household- and environmental-risk factors were estimated using mixed effects logistic regression, accounting for clustering at the household, village and cluster level (ie. random intercepts were included for these levels). Associations between STH intensity (eggs per gram) and risk factors were modelled using mixed effects negative binomial regression of egg counts, with quantity of stool assessed per sample included as an offset, again accounting for clustering at the household, village and cluster level. *A priori* interactions between age and sex were investigated, considering age as both a continuous variable, and by categorising into the three demographic groups used for stratification. We used a backwards stepwise strategy to build separate multivariable models for prevalence and intensity of both species.

## Results

### Community census profile

A total of 32,606 households were identified and approached to be censused during the census. Of these 5.5% (n = 1,789) were vacant, 2.7% (n = 877) had no household member present, 0.6% (n = 204) did not consent to the census, and 0.05% (n = 17) were unable to be censused for other reasons. Following cluster delineation, once the required number of clusters was reached, ten villages located on the periphery of the study site containing 1,969 censused households (6.6%) and 9,255 enumerated individuals (7.1%) were excluded from the main study. The resulting median cluster population was 606 households (IQR 566–791) and 2,751 individuals (IQR 2,405–3,450) (Fig 2).

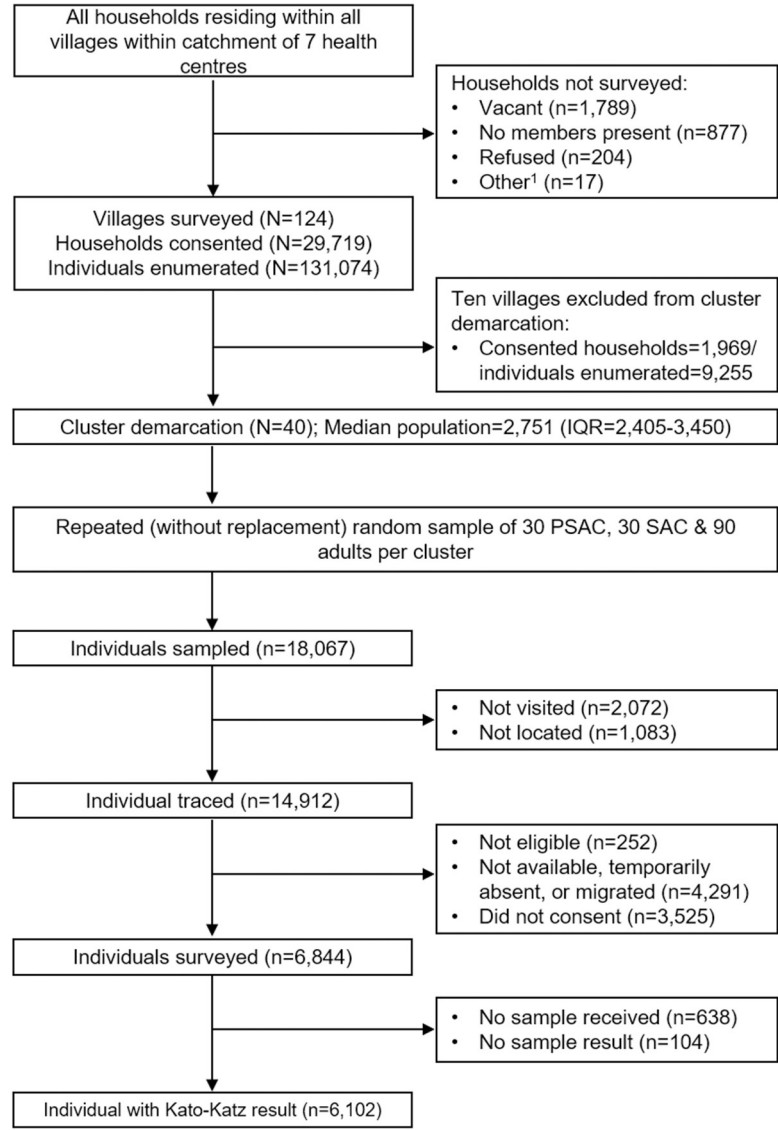

1: No adult present at household (n=9); No adult at household able to consent at that time (n=8).

**Fig 2. Study participant flow chart.**

The demographic profile of the census was highly skewed towards younger age groups, with 48.8% (n = 59,475) aged under 15 years, and females comprising 52.8% (n = 64,333) of the enumerated population (Table 1). Overall, participation in education was high, with 66.7% of males and 70.1% of females 5 years of age (one year before the primary entry age) reporting enrolment in organised learning, and 93.7% of males and 94.2% females aged 8 years (official age of Standard 2/3) reporting school enrolment. However, when disaggregated by wealth, participation in learning at all ages, and the proportion of adults completing primary education, was consistently lowest in the poorest quintile relative to least poor. Current participation in education showed broad parity by sex between the ages of 5–14, although educational attainment of adult females was lower relative to adult men.

The median number of residents per household was four (IQR: 3–6). Most households reported Islam as the primary religion practised (94.5%, n = 26,211) and Chiyao as the primary language of communication (95.3%, n = 26,449). Reported access to basic sanitation and protected water was high overall, with 93.1% of households reporting use of an improved toilet facility and 98.4% of households reporting use of an improved water source as the primary source of drinking water, although households in the least poor quintile consistently reported greater access to improved levels of sanitation facility relative to those in the poorest quintile.

## Parasitological survey enrolment

Of the 14,912 individuals approached to participate in the parasitological survey, 69.5% (n = 10,369) were present in the household and eligible. The primary reasons for unavailability were temporary absence from the area (16%, n = 2,390), permanent migration away from the area (6.6%, n = 985) or absence on the day of the survey (6.1%, n = 916). Two-thirds of present and eligible sampled participants consented to the survey (66%, n = 6,844). Of these, 6,102 (89.2%) subsequently provided a stool specimen that was assessed by Kato-Katz (Fig 2).

Overall, the individual demographic profile of parasitological survey participants showed over-representation of males in PSAC and SAC age groups and under-representation of males in adults, relative to females (Fig 3). However, the majority of other individual-level characteristics (e.g. participation in education, frequency of migration) and household-level characteristics (household demographics; dwelling materials, assets and utilities; and access to water and sanitation facilities) were broadly similar to the censused population, including when disaggregated by sex and wealth (S1 Table).

## STH infection and intensity by species

The prevalence of any STH infection across the survey population was 7.4% (1.5% in PSAC, 4.6% in SAC, and 11.2% in adults) resulting in an age-cluster weighted prevalence of 7.8% (95% C.I. 7.0%-8.6%). Hookworm was the predominant STH species (age-cluster weighted prevalence 7.5% (95% C.I. 6.7%-8.3%)) detected in all study clusters and in 83% of villages. Conversely, age and cluster-population weighted prevalence of *A. lumbricoides* and *T. trichiura* was very low (<0.1% and <0.3% respectively); with *T. trichiura* identified in 12 study clusters and *A. lumbricoides* in only 3 clusters. All STH infections were of light intensity class, with an arithmetic mean intensity of hookworm infection of 36 epg (SD = 461).

## Demographic infection profile

Examination of the age-sex profile showed that hookworm infection prevalence increased with age across both sexes, but was lower overall in females, particularly in women under the age of 35 years (Fig 4). Infection intensity remained constant until later older ages, increasing at ages 60 and 70 years in men and women respectively.

**Table 1. Individual and household-level characteristics of (i) community census participants registered in delineated study clusters in total, disaggregated by socio-economic status, and disaggregated by sex; and (ii) parasitological survey participants in total; in Namwera, Mangochi district, Malawi in 2018.**

| | Community census participants registered in delineated study clusters; % (n) | | | | | Parasitological survey participants; % (n) |
|---|---|---|---|---|---|---|
| | Total: | Q1: (Poorest) | Q5: (Least poor) | Male | Female | Total: |
| **Individual demographic profile:** | | | | | | |
| **Age group (years):** | | | | | | |
| <1 year | 3.6 (4368) | 3.9 (863) | 3 (823) | 3.7 (2121) | 3.5 (2247) | - |
| 1–4 years | 14.3 (17455) | 16 (3520) | 12 (3265) | 14.9 (8577) | 13.8 (8876) | 21.8 (1329) |
| 5–14 years | 30.9 (37652) | 30.6 (6728) | 32 (8732) | 32.7 (18817) | 29.3 (18832) | 25.3 (1541) |
| ≥15 years | 51 (62161) | 49.4 (10852) | 52.9 (14422) | 48.4 (27838) | 53.4 (34323) | 52.9 (3228) |
| Age unknown | 0.2 (183) | 0.1 (22) | 0.1 (36) | 0.2 (128) | 0.1 (55) | 0.07 (4) |
| **Individual stayed in household the majority of days in 6 months prior to census survey** | 96.2 (117189) | 97.3 (21386) | 95.6 (26071) | 94.6 (54400) | 97.6 (62784) | 98.4 (6007) |
| **Individual slept at household night prior to census survey** | 94.5 (115078) | 95.3 (20956) | 93.6 (25519) | 90.9 (52228) | 97.7 (62845) | 98.2 (5989) |
| **Age specific enrolment rate:** | | | | | | |
| Age 5 years (OA 1 year before Std 1) | 68.4 (2981) | 60.4 (530) | 77.3 (686) | 66.7 (1437) | 70.1 (1544) | 64.2 (154) |
| Age 8 years (OA Std 2/3) | 94 (3444) | 90.9 (628) | 96.6 (791) | 93.7 (1670) | 94.2 (1773) | 96.8 (152) |
| Age 11 years (OA Std 5/6) | 96 (3024) | 94.1 (481) | 97.8 (750) | 95.7 (1541) | 96.4 (1483) | 94 (126) |
| Age 14 years (OA Std 8) | 89.1 (2774) | 84.8 (441) | 92.2 (707) | 90.6 (1449) | 87.4 (1325) | 91.3 (94) |
| **Highest level of education (age≥15 years):** | | | | | | |
| No formal education | 34.9 (21350) | 46.2 (4920) | 23.7 (3382) | 25.2 (6825) | 42.6 (14525) | 37.3 (1197) |
| Primary incomplete | 49.1 (30083) | 46.5 (4947) | 45.7 (6504) | 53.1 (14395) | 46 (15688) | 50.4 (1620) |
| Primary complete or higher | 13 (7954) | 4.6 (484) | 26.8 (3817) | 16 (4351) | 10.6 (3603) | 10.3 (332) |
| Highest education level unknown | 3 (1862) | 2.7 (291) | 3.8 (541) | 5.7 (1550) | 0.9 (312) | 2 (64) |
| **Household demographic profile:** | | | | | | |
| **Median household size (range)** | 4 (1–17) | 4 (1–14) | 5 (1–16) | - | - | - |
| **Household stayed less than 5 years** | 48.0 (13326) | 39.7 (2204) | 50.9 (2780) | - | - | 43.0 (1914) |
| **Primary language spoken is Chiyao** | 95.3 (26449) | 99.1 (5499) | 86.5 (4726) | - | - | 95.9 (4264) |
| **Primary religion practiced is Islam** | 94.5 (26211) | 98.0 (5438) | 86.5 (4726) | - | - | 94.9 (4219) |
| **Household materials, assets & utilities:** | | | | | | |
| **Dwelling has floor of natural material** | 79.6 (22077) | 99.6 (5530) | 40.8 (2229) | - | - | 81.0 (3604) |
| **Household has any livestock** | 35.1 (9729) | 21.3 (1181) | 51.1 (2789) | - | - | 39.8 (1768) |
| **Household has electricity** | 4.8 (1327) | 0 (0) | 19.0 (1040) | - | - | 3.5 (157) |
| **Household water and sanitation:** | | | | | | |
| **Household toilet facility:** | | | | | | |
| Basic | 68.0 (18882) | 58.7 (3260) | 78.1 (4263) | - | - | 72.3 (3213) |
| Limited | 25.1 (6970) | 30.5 (1691) | 19.4 (1060) | - | - | 21.5 (955) |
| Unimproved | 4.3 (1193) | 5.9 (327) | 2.1 (113) | - | - | 4.5 (199) |
| Open defecation | 2.5 (705) | 4.9 (272) | 0.5 (25) | - | - | 1.8 (80) |
| **Household water source:** | | | | | | |
| Basic | 74.9 (20795) | 72.1 (4000) | 77.9 (4256) | - | - | 72.4 (3221) |
| Limited | 23.4 (6483) | 26.0 (1444) | 21.0 (1149) | - | - | 24.2 (1077) |
| Unimproved | 1.5 (411) | 1.7 (93) | 0.9 (47) | - | - | 1.3 (58) |
| Surface water | 0.2 (61) | 0.2 (13) | 0.2 (9) | - | - | 0.2 (7) |
| Unknown | - | - | - | - | - | 1.9 (84) |

Abbreviations: OA Std = Official age for standard (grade level).

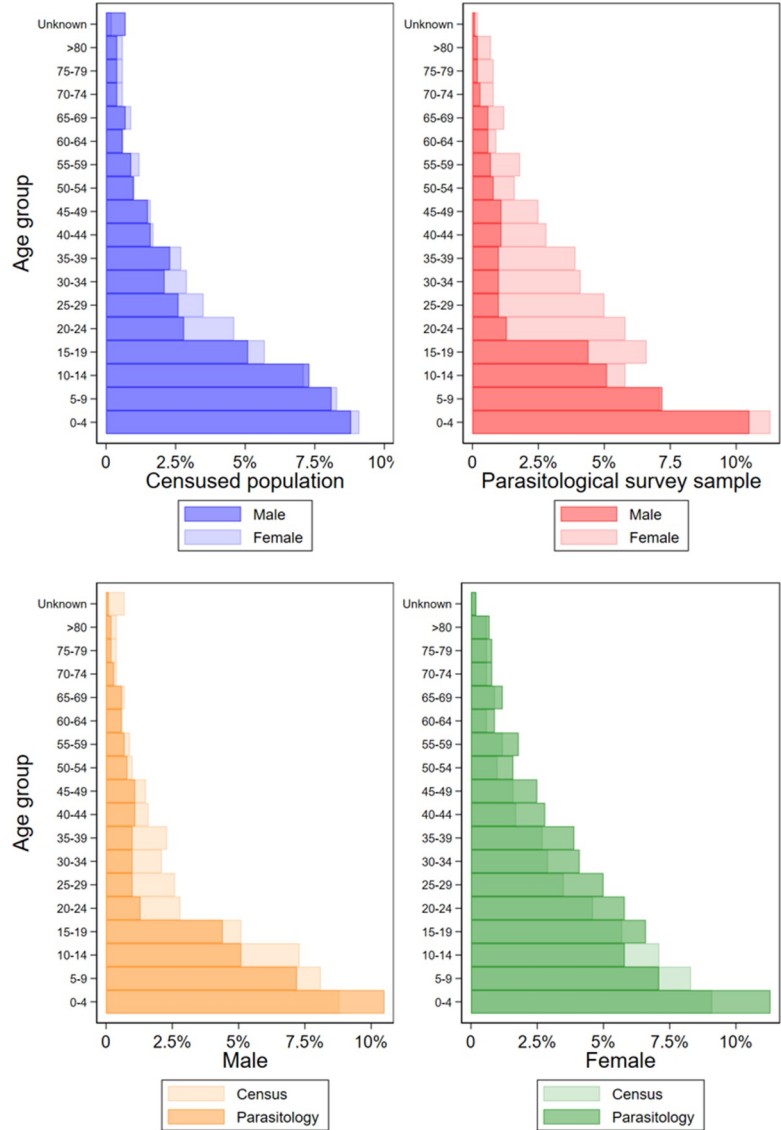

**Fig 3. Population structure of census and participation in parasitological survey by age group and sex.**

## Predictors of hookworm infection

In univariable analysis the majority of surveyed individual, household and environmental factors investigated were associated with hookworm infection, except for access to sanitation in the household and other WASH-related factors (Table 2). Adjusting for covariates in a multivariable model, many of these associations remained, with females and younger age groups consistently exhibiting lower odds of infection. There was evidence of interaction between age and sex, with the difference in odds of infection by sex being most pronounced among adults. Individuals from the least poor households (as assessed by household material and asset index) had lower odds of hookworm infection, which was also detected to a lesser extent in households where a formal occupation was the source of household income. No association was observed with access to sanitation facilities, handwashing facilities, or having to share sanitation facilities. Presence of man-made flooring materials in the household was associated with

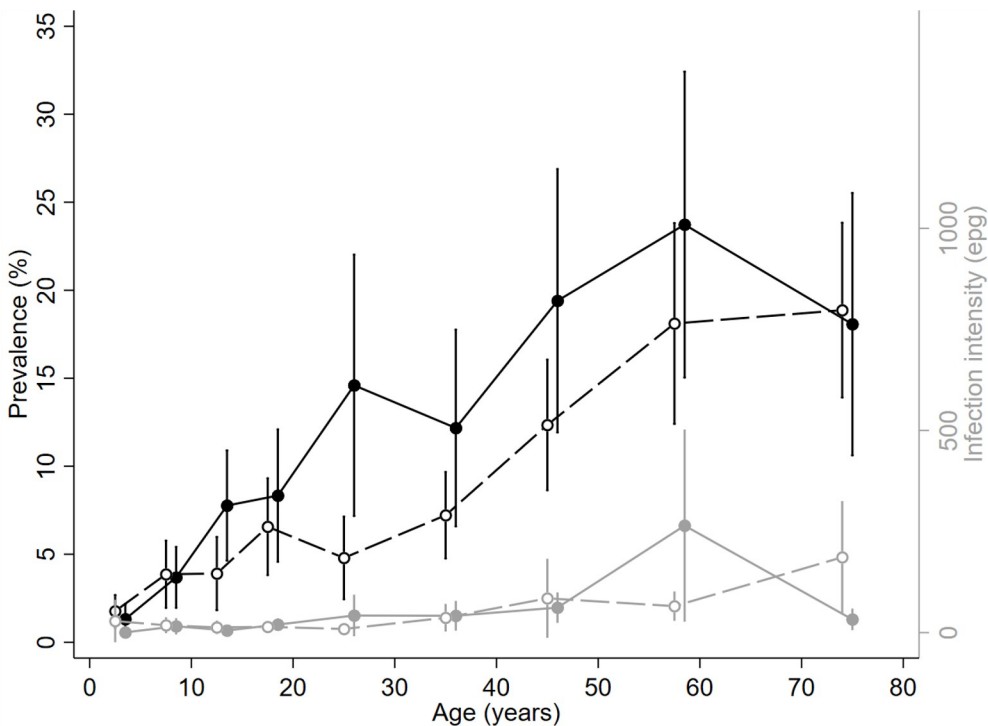

**Fig 4. Hookworm age-sex-infection profiles.** Prevalence (black) and intensity (grey) of hookworm infection by age (years) and sex (males= solid line and circles, females= dashed lines and empty circles).

lower odds of hookworm infection. Currently attending school and living with a household member who has completed at least primary school education were independently associated with reduced odds of infection, relative to not attending school or living in households where adults had no education, respectively. We observed higher odds of hookworm infection in dry sub-humid areas (versus semi-arid) and with increasing elevation and topsoil sand fraction. We also observed lower odds of hookworm infection in urban households (versus peri-urban and rural).

## Predictors of hookworm intensity

Hookworm infection intensity was associated with many of the same factors as presence of infection, including older age, although there was no evidence of a difference by sex (Table 3). The relationship between hookworm infection and poverty was similar, with those in the least poor households having significantly lower intensity of infection. The protective effect of education also persisted albeit to a lesser extent, with those currently in education having significantly lower intensity of infection. No evidence of an association was observed between access to sanitation, handwashing facilities, or use of shared sanitation with intensity of infection. Urban residence was also not associated with infection intensity, but elevation and sand fraction were strongly associated with lower infection intensity.

Comparing the model random effects at the cluster, village and household level highlights the extent by which both presence and intensity of infection were highly clustered within households in this site, even after adjusting for individual and household level factors.

## *Schistosoma mansoni* and other detected infections

The prevalence of any *S. mansoni* infection across the survey population was 1.7% (0.5% in PSAC, 0.7% in SAC and 2.6% in adults), resulting in an age and cluster-population weighted

**Table 2. Predictors of presence of hookworm infection amongst preschool-age children, school-age children, and adults in Namwera, Mangochi district, Malawi in 2018.**

| Characteristic | | % (N) of participants[1] | % (n) of participants with hookworm infection | Univariable analysis[2] | | Multivariable analysis[2,3] | |
|---|---|---|---|---|---|---|---|
| | | | | OR (95% CI) | P value | Adjusted OR (95% CI) | P value |
| **Individual factors** | | | | | | | |
| **Sex:** | | | | | | | |
| Male | | 37.1 (2263) | 7.7 (175) | - | - | - | - |
| Female | | 62.9 (3839) | 6.8 (261) | - | - | - | - |
| **Age group:** | | | | | | | |
| 1–4 years | | 21.8 (1329) | 1.4 (18) | - | - | - | - |
| 5–14 years | | 25.3 (1541) | 4.2 (65) | - | - | - | - |
| ≥15 years | | 52.9 (3228) | 10.9 (352) | - | - | - | - |
| **Effect of age by sex:** | | | | | | | |
| Male | 1–4 years | 28.3 (640) | 1.1 (7) | 1 | - | 1 | - |
| | 5–14 years | 33.3 (753) | 4.7 (35) | 5.55 (2.23–13.81) | - | 9.21 (3.48–24.35) | |
| | ≥15 years | 38.4 (870) | 15.3 (133) | 28.23 (11.08–71.95) | <0.001 | 28.18 (11.06–71.76) | <0.001 |
| Female | 1–4 years | 18.0 (689) | 1.6 (11) | 1 | - | 1 | - |
| | 5–14 years | 20.5 (788) | 3.8 (30) | 2.74 (1.24–6.03) | - | 4.58 (1.95–10.75) | |
| | ≥15 years | 61.5 (2362) | 9.3 (220) | 8.61 (4.13–17.95) | <0.001 | 7.62 (3.66–15.84) | <0.001 |
| **Effect of sex by age:** | | | | | | | |
| 1–4 years | Male | 48.2 (640) | 1.1 (7) | 1 | - | 1 | - |
| | Female | 51.8 (689) | 1.6 (11) | 1.57 (0.56–4.42) | 0.40 | 1.51 (0.53–4.26) | 0.43 |
| 5–14 years | Male | 48.9 (753) | 4.7 (35) | 1 | - | 1 | - |
| | Female | 51.1 (788) | 3.8 (30) | 0.77 (0.43–1.39) | 0.39 | 0.75 (0.42–1.36) | 0.34 |
| ≥15 years | Male | 26.9 (870) | 15.3 (133) | 1 | - | 1 | - |
| | Female | 73.1 (2362) | 9.3 (220) | 0.47 (0.34–0.65) | <0.001 | 0.41 (0.29–0.57) | <0.001 |
| **Currently in education:** | | | | | | | |
| No | | 65 (3965) | 8.9 (353) | 1 | - | 1 | - |
| Yes | | 35 (2137) | 3.9 (83) | 0.34 (0.24–0.48) | <0.001 | 0.44 (0.29–0.68) | <0.001 |
| **Dewormed within past 12 months:** | | | | | | | |
| No | | 49.3 (2928) | 8.9 (261) | 1 | - | - | - |
| Yes | | 50.7 (3011) | 5.4 (162) | 0.49 (0.37–0.66) | <0.001 | - | - |
| **Household factors** | | | | | | | |
| **Access to sanitation:** | | | | | | | |
| Open defecation | | 1.7 (104) | 7.7 (8) | 1 | - | - | - |
| Unimproved | | 273 (4.5) | 7 (19) | 1.07 (0.36–3.19) | - | - | - |
| Limited | | 1240 (20.3) | 7.7 (95) | 0.94 (0.36–2.43) | - | - | - |
| Basic | | 4485 (73.5) | 7 (314) | 0.73 (0.29–1.85) | 0.30 | - | - |
| **Sanitation facility is shared:** | | | | | | | |
| No | | 78.7 (4805) | 7 (338) | 1 | - | - | - |
| Yes | | 21.3 (1297) | 7.6 (98) | 1.24 (0.92–1.68) | 0.16 | - | - |
| **Access to handwashing facility:** | | | | | | | |
| No facility | | 25.6 (1520) | 6.4 (97) | 1 | - | - | - |
| Limited | | 69.0 (4092) | 7.6 (312) | 1.23 (0.90–1.69) | - | - | - |
| Basic | | 5.3 (317) | 4.1 (13) | 0.61 (0.29–1.28) | 0.08 | - | - |
| **Household flooring:** | | | | | | | |
| Natural materials | | 81.1 (4940) | 383 (7.8) | 1 | - | - | - |
| Man-made materials | | 18.9 (1155) | 51 (4.4) | 0.59 (0.41–0.86) | 0.005 | - | - |
| **Socio-economic status:** | | | | | | | |
| Poorest (Q1) | | 18.7 (1138) | 11.2 (127) | 1 | - | 1 | - |
| Q2 | | 20.2 (1232) | 7.8 (96) | 0.64 (0.44–0.93) | - | 0.63 (0.43–0.92) | - |

*(Continued)*

**Table 2.** (Continued)

| Characteristic | % (N) of participants[1] | % (n) of participants with hookworm infection | Univariable analysis[2] | | Multivariable analysis[2,3] | |
|---|---|---|---|---|---|---|
| | | | OR (95% CI) | *P* value | Adjusted OR (95% CI) | *P* value |
| Q3 | 18.8 (1149) | 6.2 (71) | 0.54 (0.36–0.81) | - | 0.55 (0.37–0.84) | - |
| Q4 | 21 (1282) | 6.9 (88) | 0.61 (0.42–0.89) | - | 0.60 (0.40–0.89) | - |
| Least poor (Q5) | 21.3 (1301) | 4.2 (54) | 0.34 (0.22–0.53) | <0.001 | 0.37 (0.23–0.59) | 0.001 |
| **Source of household income:** | | | | | | |
| *Ganyu*[4] only | 38.4 (2170) | 8 (174) | 0.93 (0.58–1.51) | - | - | - |
| Farming | 33.5 (1893) | 7.7 (146) | 1.02 (0.62–1.67) | - | - | - |
| Other informal | 24.3 (1376) | 5.3 (73) | 0.60 (0.35–1.02) | - | - | - |
| Formal | 3.8 (215) | 2.8 (6) | 0.36 (0.13–1.02) | 0.02 | - | - |
| **Highest education level of any adult household member:** | | | | | | |
| No education | 25.0 (1515) | 9.2 (1515) | 1 | - | 1 | - |
| Primary incomplete | 56.5 (3429) | 7.2 (3429) | 0.71 (0.53–0.94) | - | 0.75 (0.56–1.01) | - |
| Primary complete or higher | 18.6 (1128) | 4.3 (49) | 0.43 (0.28–0.67) | <0.001 | 0.50 (0.32–0.78) | 0.009 |
| **Household size:** | | | | | | |
| *# household members* | - | - | 0.89 (0.83–0.94) | <0.001 | - | - |
| **Environmental factors** | | | | | | |
| **Aridity:** | | | | | | |
| Semi-arid | 37 (2259) | 4.9 (111) | 1 | - | - | - |
| Dry sub-humid | 63 (3842) | 8.5 (325) | 1.78 (1.15–2.75) | 0.009 | - | - |
| **Elevation:** | | | | | | |
| *Mean, SD* | - | - | 1.005 (1.001–1.007) | 0.001 | - | - |
| **Sand fraction:** | | | | | | |
| *Mean, SD* | - | - | 0.96 (0.93–0.98) | 0.004 | 0.94 (0.92–0.98) | 0.001 |
| **NDVI (Greenness proxy):** | | | | | | |
| *Mean, SD* | - | - | 21.81 (0.51–929.70) | 0.11 | 81.63 (1.65–4027.05) | 0.027 |
| **Urbanisation** | | | | | | |
| Rural | 7.9 (482) | 8.7 (42) | 1 | - | - | - |
| Peri-urban | 55 (3355) | 8.2 (274) | 0.81 (0.51–1.29) | - | - | - |
| Urban | 37 (2265) | 5.3 (120) | 0.48 (0.28–0.81) | 0.003 | - | - |

[1]6102 observations with Kato-katz result included. All variables have complete data, with the exception of age group (n = 6098), dewormed within past 12 months (n = 5939), access to handwashing facility (n = 5929), household flooring (n = 6095), source of household income (n = 5654) and highest education level of any adult household member (n = 6072).

[2]Mixed effects logistic regression (MELR) accounting for clustering at the household, village and cluster level. Reference category is indicated by a value of [1].

[3]6070 observations included in fully adjusted MELR model.

[4]Casual off-own-farm labour (e.g. weeding or ridging).

Acronyms: CI = Confidence interval, EPG = Eggs per gram, OR = Odds ratio, SD = Standard deviation.

prevalence of 1.6% (95% CI 1.3–2.0%). Infection with *S. mansoni* was detected in 32 of the 40 study clusters, with similar infection prevalence observed in males (1.5%) and females (1.7%). Other infections detected were *Enterobius vermicularis* (0.6%, n = 37, in 25 clusters) and *Hymenolepis nana* (<0.1%, n = 4, observed in 3 clusters).

## Discussion

This community-based study of more than 6,000 individuals in southern Malawi provides a detailed picture of contemporary demographics and community-level epidemiology of STH in

**Table 3. Predictors of intensity of hookworm infection amongst preschool-age children, school-age children, and adults in Namwera, Mangochi district, Malawi in 2018.**

| Characteristic | Mean hookworm EPG (SD)[1] | Univariable analysis[2] | | Multivariable analysis[2,3] | |
|---|---|---|---|---|---|
| | | IRR (95% CI) | P value | Multivariable IRR (95% CI) | P value |
| **Individual factors** | | | | | |
| **Sex:** | | | | | |
| Male | 32 (345) | 1 | - | - | - |
| Female | 38 (518) | 1.04 (0.62–1.76) | 0.87 | - | - |
| **Age group:** | | | | | |
| <1 year | 16 (548) | 1 | - | 1 | - |
| 1–4 years | 14 (180) | 11.21 (4.35–28.90) | - | 21.56 (8.64–53.82) | - |
| 5–14 years | 54 (512) | 148.53 (60.23–366.29) | <0.001 | 25.40 (12.66–50.98) | <0.001 |
| **Currently in education:** | | | | | |
| No | 50 (561) | 1 | - | 1 | - |
| Yes | 10 (152) | 0.12 (0.06–0.21) | <0.001 | 0.23 (0.11–0.50) | <0.001 |
| **Dewormed within past 12 months:** | | | | | |
| No | 48 (513) | 1 | - | 1 | - |
| Yes | 23 (411) | 0.27 (0.15–0.46) | <0.001 | 0.60 (0.35–1.02) | 0.006 |
| **Household factors** | | | | | |
| **Access to sanitation:** | | | | | |
| Open defecation | 30 (197) | 1 | - | - | - |
| Unimproved | 11 (73) | 0.71 (0.08–6.60) | - | - | - |
| Limited | 56 (769) | 1.13 (0.17–7.55) | - | - | - |
| Basic | 32 (354) | 0.67 (0.10–4.29) | 0.44 | - | - |
| **Sanitation facility is shared:** | | | | | |
| No | 31 (343) | 1 | - | - | - |
| Yes | 54 (751) | 1.58 (0.85–2.94) | 0.15 | - | - |
| **Access to handwashing facility:** | | | | | |
| No facility | 39 (379) | 1 | - | - | - |
| Limited | 33 (468) | 0.79 (0.42–1.48) | - | - | - |
| Basic | 51 (720) | 0.41 (0.10–1.59) | 0.42 | - | - |
| **Household flooring:** | | | | | |
| Natural materials | 38 (456) | 1 | - | - | - |
| Man-made materials | 26 (484) | 0.50 (0.24–1.03) | 0.06 | - | - |
| **Socio-economic status:** | | | | | |
| Poorest (Q1) | 72 (782) | 1 | - | 1 | - |
| Q2 | 30 (245) | 0.47 (0.22–1.02) | - | 0.37 (0.18–0.79) | - |
| Q3 | 33 (432) | 0.51 (0.22–1.18) | - | 0.69 (0.32–1.52) | - |
| Q4 | 37 (432) | 0.57 (0.25–1.27) | - | 0.62 (0.29–1.34) | - |
| Least poor (Q5) | 10 (77) | 0.19 (0.08–0.45) | 0.005 | 0.29 (0.13–0.65) | 0.013 |
| **Source of household income:** | | | | | |
| *Ganyu*[4] only | 50 (604) | 1.07 (0.39–2.96) | - | - | - |
| Farming | 33 (329) | 1.15 (0.41–3.20) | - | - | - |
| Other informal | 27 (329) | 0.42 (0.14–1.26) | - | - | - |
| Formal | 27 (461) | 0.19 (0.03–1.20) | 0.02 | - | - |
| **Highest education level of any adult household member:** | | | | | |
| No education | 45 (416) | 1 | - | 1 | - |
| Primary incomplete | 34 (467) | 0.48 (0.26–0.87) | - | 0.74 (0.42–1.30) | - |
| Primary complete or higher | 29 (507) | 0.26 (0.11–0.60) | 0.005 | 0.46 (0.21–0.97) | 0.12 |

*(Continued)*

**Table 3.** (Continued)

| Characteristic | Mean hookworm EPG (SD)[1] | Univariable analysis[2] | | Multivariable analysis[2,3] | |
|---|---|---|---|---|---|
| | | IRR (95% CI) | *P* value | Multivariable IRR (95% CI) | *P* value |
| **Household size:** | | | | | |
| *# household members* | - | 0.83 (0.74–0.92) | 0.001 | - | - |
| **Environmental factors** | | | | | |
| **Aridity:** | | | | | |
| Semi-arid | 20 (300) | 1 | - | - | - |
| Dry sub-humid | 45 (534) | 2.52 (1.09–5.82) | 0.03 | - | - |
| **Elevation:** | | | | | |
| *Mean, SD* | - | 1.007 (1.001–1.011) | 0.014 | - | - |
| **Sand fraction:** | | | | | |
| *Mean, SD* | - | 0.91 (0.86–0.97) | 0.002 | 0.91 (0.85–0.96) | 0.001 |
| **Population density within 1km** | | | | | |
| <50 | 7 (54) | 1 | - | - | - |
| 50–249 | 41 (456) | 7.22 (1.47–35.38) | - | - | - |
| >249 | 33 (475) | 4.44 (0.90–21.98) | 0.03 | - | - |

[1]6102 observations with Kato-katz result included. All variables have complete data, with the exception of age group (n = 6098), dewormed within past 12 months (n = 5939), access to handwashing facility (n = 5929), household flooring (n = 6095), source of household income (n = 5654) and highest education level of any adult household member (n = 6072).

[2]Mixed effects negative binomial regression of egg counts, with quantity of stool assessed per sample included as an offset, accounting for clustering at the household, village and cluster level. Reference category is indicated by a value of [1].

[3]5885 observations included in fully adjusted mixed effects negative binomial regression model.

[4]Casual off-own-farm labour (e.g. weeding or ridging).

Acronyms: CI = Confidence interval, EPG = Eggs per gram, IRR = Incidence rate ratio, SD = Standard deviation.

2018, in a context of MDA with albendazole delivered consistently to targeted demographic groups through multiple platforms for more than a decade. The findings presented here are likely to be broadly comparable to settings with similar socio-demographic and environmental profiles that have successfully implemented and sustained routinely preventive chemotherapy through school-based deworming and other routine delivery platforms.

The observed prevalence of STH infection in pre-school (aged 1–4 years) and school-age (aged 5–14 years) children was low (1.4% and 4.2% respectively) consisting of predominately hookworm infection (>99%). Published literature on the prevalence of STH infection in children in this region is very limited, but broadly describes a setting where STH infection, generally starting from a moderate prevalence, has declined to very low levels during the past two decades. School-based surveys conducted in 1999 in the north of Malawi with school-age children reported a hookworm prevalence of 64% [24], with subsequent research conducted between 2000–2002 describing an STH prevalence in children aged 3–14 of 3.6% and 16.5% in rural and urban communities respectively in southern Malawi [25] and a national school-based survey reporting an STH prevalence of 1.8% (95% CI: 0.6–3.1) [26]. Following scale-up of school-based deworming, a 2008 facility-based study of severe anaemia in pre-school age children reported a hookworm prevalence of 2.5% in community-based controls [27] and 2011 community-based survey reported an STH prevalence of 0.3–3.8% [11]. In contrast, a school-based survey of 7,491 schoolchildren in the bordering Niassa province of Mozambique, where school-based deworming had not been routinely conducted, reported a mean STH prevalence of 51.4% (95% CI: 12.3–78.8) as recently as 2005–2007 [28].

The observed prevalence of 10.9% STH infection in adults (aged $\geq$15 years) in this setting was substantially higher than in children, and hookworm species were most common in adults (>99%). The only published literature of STH infection in Malawian adults that we are aware of is a parasitological survey of 848 pregnant women aged 17–23 years conducted between 2002–2004 in the neighbouring Machinga district which supports a similar pattern of declining prevalence; reporting a hookworm prevalence of 14.4% with 95% of infections being low intensity, and few (<0.5%) non-hookworm infections [29]. While the increased prevalence of hookworm infection with age has long been recognised [30] our survey confirms the result of an age-structured cross-sectional survey conducted in Kenya [13], a setting that has similarly delivered wide-scale MDA for STH. In addition to more limited surveys conducted in Kenya [31,32], these results confirm that whilst deworming programmes may have made a substantial impact on hookworm infection prevalence in children, adults continue to remain at increased risk of infection and importantly, as a reservoir of STH infection in these communities.

As morbidity attributable to infection with STH, including the clinical sequelae of iron-deficiency anaemia, is most prominent when worm burden is relatively high, global targets for STH control continue to prioritise reduction in morbidity rather than infection [33]. As no MHI STH infections were detected in this survey, our results demonstrate that this region of Malawi has already achieved the WHO 2030 milestone of elimination of STH as a public health problem (<2% of MHI STH infection) [34]. This finding is notable as it has been achieved despite the higher prevalence of STH observed in adult males, who have not been routinely targeted with anthelmintic treatment following the cessation of MDA for LF. However, in this setting of high vulnerability to many other risk factors for anaemia, including chronic nutritional deficiencies and the consequences of other infectious diseases, the prevalence of moderate or severe anaemia remains almost twice the national average, with one in eight women aged 15–49 having moderate or severe anaemia [7]. As such, while approaches to increase treatment coverage of adults in this setting would likely be an effective approach to further reducing the prevalence of STH, in this respect there may be limited clinical benefit.

Across a range of settings, access to sanitation is strongly associated with reduced risk of STH transmission [35]. Despite this, sanitation has been highlighted as an inadequately emphasised component of many STH control strategies [36–38]. The prioritisation of routine deworming over improvements in sanitation is in part justified on the rationale that the resources required to do so are rarely available in settings where STH are endemic [39]. In this study, we observe a high level of reported access to improved sanitation at home, confirming evidence that access to sanitation has continued to grow and be sustained in Malawi, with 71% of households in 1995 [40] and 85% of households in 2006 [41] in Mangochi district with reported access to improved sanitation. Given the high levels of coverage, it is perhaps unsurprising that we did not observe an association between access to sanitation and STH infection, and it is highly likely that this sustained level of sanitation has played a major role in sustaining the reductions in STH infection achieved through routine deworming.

In this setting of predominantly agricultural livelihoods, we suggest that access to sanitation when away from the household may play a relatively important role in continuing to drive transmission in this setting. In contrast to household-level access, 23% of participants reported open defecation when at work, and moderate evidence of lower levels of infection in those from households whose primary occupation is formal where access to sanitation would plausibly be higher. Research from other rural settings in SSA has observed no protective association of community-level sanitation on hookworm infection even at high levels [42,43]. Forthcoming research will explore how risk of STH infection changes relative to access to sanitation at the local and school levels in this setting.

This study confirms a number of other well-established risk factors for STH infection and intensity, including the strong relationship between STH and poverty (as defined by asset and household-materials based index). This observation underlines the persistent nature of NTDs to disproportionately affect the most marginalised and vulnerable members of society. Interestingly, we observe a persistent protective effect of higher levels of household education and urbanisation, suggesting these may act along pathways independent of wealth to reduce the risk of STH infection. Furthermore, we observe the protective effect of current school enrolment, likely demonstrating the success of school-based deworming in this setting, and we confirm known environmental risk factors for STH including household crowding, soil sand fraction and aridity.

While this study was able to overcome the biases inherent in many community-representative surveys through the use of a recent population census as a sampling frame, a major limitation of the survey is bias in the demographic profile of the final enrolled sample when compared to the census. The final sample consisted of disproportionately more (adult) women, due to the replacement of male participants who were either unavailable or refused to participate in the survey, and subsequent skew towards those more likely to be present at the time visit. In addition, we also note that the final sample is over representative of non-migrants and children not attending school, likely due in part to the survey characteristic of visiting household during the daytime.

While the Kato-Katz method remains the primary diagnostic tool for detection of STH eggs in both routine monitoring and large-scale research studies, the relatively poor sensitivity of the method, particularly in settings of low prevalence, is widely recognised as a limitation of the method [44] although multiple slides or consecutive stool samples have been demonstrated to improve sensitivity [45,46]. Despite these limitations, novel molecular approaches, such as the use of quantitative polymerase chain reaction (qPCR) generally remain inappropriate for the monitoring or evaluation of STH control interventions [47], with recent evidence suggesting that qPCR is only more sensitive than Kato-katz for infections of very low intensity [48]. On this basis, Kato-katz remains sufficiently sensitive to assess the broad distribution of STH infections, and their associated risk factors, in settings with active ongoing STH transmission such as Malawi.

In conclusion, the results of this study demonstrate that concerted efforts to control soil-transmitted helminths through MDA with albendazole, facilitated by sustained access to sanitation, has successfully achieved what was intended—reducing the profile of this disease to a very low prevalence and intensity in those at greatest risk of morbidity—while transmission continues at low levels amongst adults and marginalised communities. This raises a challenging decision for policy makers and researchers alike: whether control programmes should continue to try and sustain the elimination of STH morbidity, or pivot towards expanding coverage to the entire community with the target of elimination of STH infection [49]. Building on evidence from Kenya in which community-wide MDA was more effective and feasible approach to reducing the prevalence and intensity of hookworm relative to school-based deworming over the course of two years [50,51], the Deworm3 trial [15,52] which this study sits within, aims to evaluate whether such a strategy could be effective, feasible and cost-effective [53].

## Supporting information

**S1 STOBE Checklist. STROBE checklist.**
(DOC)

**S1 Table. Individual and household-level characteristics of parasitological survey participants in total, disaggregated by socio-economic status, and disaggregated by sex; in**

**Namwera, Mangochi district, Malawi in 2018.**
(DOCX)

## Acknowledgments

We sincerely thank all those who contributed to this study, with special recognition of the individual contributions of Chikondi Chikotichalera, Fraser Chisale and Cidreck Nkomela.

We thank all community members who participated in this study, and we gratefully acknowledge the support of the Traditional Authorities of Bwananyambi, Chowe, Jalasi and Katuli, and the Area Development Committees of Bwanayambi, Majuni, Malombola and Mandimba; with whose permission this study was conducted.

We thank the team of field enumerators and laboratory technicians who contributed directly to data collection; and the drivers, health surveillance assistants, laboratory cleaners and village volunteers who tirelessly supported them.

We thank those at Mangochi DHO (Dr Henry Chibowa, Dr Kondwani Mamba and Dr Wayne Peno) and Namwera zone (Alinafe Hauya, Cornelius Kunkeyani, Chifundo Manong'a, Atusaye Mbisa and Francis Mwanoka) for their support and supervision during this study.

Lastly, we thank those at Blantyre Institute for Community Outreach (Nellie Chatsika, David Chinyanya, Roselyn Hara, Florence Kalua, Wongani Lungu, Lifa Mandala, Limbani Mitengo, Maghanoghano Mpata, Christopher Phiri, Ranneck Singano and Rose Wilson), the College of Medicine (Dr. Sarah Burr and Harry Meleke), Kenya Medical Research Institute (Paul Gichuki and Anne Njoka), LSHTM (Jessie Hammon, John Hart and Eleanor Martins) and Natural History Museum (Leanne Doran, Iain Gardiner, Dr Tim Littlewood and Elodie Yard) who contributed to the planning, coordination, and implementation of this study.

## Author Contributions

**Conceptualization:** Kristjana Ásbjörnsdóttir, Katherine E. Halliday, Judd L. Walson, Robin L. Bailey, Khumbo Kalua, Rachel L. Pullan.

**Data curation:** Stefan Witek-McManus, Alvin B. Chisambi, William E. Oswald, David S. Kennedy, Joseph W. S. Timothy, Sean R. Galagan, Mira Emmanuel-Fabula.

**Formal analysis:** Stefan Witek-McManus, Alvin B. Chisambi, William E. Oswald, David S. Kennedy, Sean R. Galagan, Rachel L. Pullan.

**Funding acquisition:** Judd L. Walson, Robin L. Bailey, Khumbo Kalua, Rachel L. Pullan.

**Investigation:** Stefan Witek-McManus, James Simwanza, Alvin B. Chisambi, Stella Kepha, Zachariah Kamwendo, Alfred Mbwinja, Lyson Samikwa.

**Methodology:** Stefan Witek-McManus, James Simwanza, Alvin B. Chisambi, Stella Kepha, Zachariah Kamwendo, William E. Oswald, Fabian Schaer, Kristjana Ásbjörnsdóttir, Katherine E. Halliday, Robin L. Bailey, Khumbo Kalua, Rachel L. Pullan.

**Project administration:** Stefan Witek-McManus, James Simwanza, Alvin B. Chisambi, Stella Kepha, Zachariah Kamwendo, Alfred Mbwinja, Hugo Legge, Khumbo Kalua.

**Resources:** Lazarus Juziwelo, Robin L. Bailey, Khumbo Kalua.

**Software:** William E. Oswald, David S. Kennedy, Sean R. Galagan, Mira Emmanuel-Fabula.

**Supervision:** Stella Kepha, Fabian Schaer, Kristjana Ásbjörnsdóttir, Katherine E. Halliday, Lazarus Juziwelo, Robin L. Bailey, Khumbo Kalua, Rachel L. Pullan.

**Validation:** Stella Kepha, Lyson Samikwa, Sean R. Galagan, Mira Emmanuel-Fabula.

**Visualization:** Stefan Witek-McManus, Rachel L. Pullan.

**Writing – original draft:** Stefan Witek-McManus, Rachel L. Pullan.

**Writing – review & editing:** James Simwanza, Stella Kepha, Alfred Mbwinja, William E. Oswald, David S. Kennedy, Hugo Legge, Sean R. Galagan, Fabian Schaer, Katherine E. Halliday, Judd L. Walson, Robin L. Bailey, Khumbo Kalua.

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
