## [Decision Letter · Decision Letter 0]

23 Dec 2020

Dear Mr Witek-McManus,

Thank you very much for submitting your manuscript "Epidemiology of soil-transmitted helminths following sustained implementation of routine preventive chemotherapy: demographics and baseline results of a cluster randomised trial in southern Malawi" for consideration at PLOS Neglected Tropical Diseases. As with all papers reviewed by the journal, your manuscript was reviewed by members of the editorial board and by several independent reviewers. The reviewers appreciated the attention to an important topic. Based on the reviews, we are likely to accept this manuscript for publication, providing that you modify the manuscript according to the review recommendations. 

Sincerely,

Prof. María-Gloria Basáñez, PhD, MSc

Associate Editor

Aysegul Taylan Ozkan

Deputy Editor

Reviewer's Responses to Questions

**Key Review Criteria Required for Acceptance?**

**Methods**

-Are the objectives of the study clearly articulated with a clear testable hypothesis stated?

-Is the study design appropriate to address the stated objectives?

-Is the population clearly described and appropriate for the hypothesis being tested?

-Is the sample size sufficient to ensure adequate power to address the hypothesis being tested?

-Were correct statistical analysis used to support conclusions?

-Are there concerns about ethical or regulatory requirements being met?

Reviewer #1: OK

Reviewer #2: Yes

Reviewer #3: Yes

**Results**

-Does the analysis presented match the analysis plan?

-Are the results clearly and completely presented?

-Are the figures (Tables, Images) of sufficient quality for clarity?

Reviewer #1: - I wonder why there is no P-value mentioned in the Univariable analysis for Household flooring (both in prevalence and intensity column).

Reviewer #2: Yes

Reviewer #3: Yes

**Conclusions**

-Are the conclusions supported by the data presented?

-Are the limitations of analysis clearly described?

-Do the authors discuss how these data can be helpful to advance our understanding of the topic under study?

-Is public health relevance addressed?

Reviewer #1: Clear conclusions

Reviewer #2: Yes

Reviewer #3: Yes

**Editorial and Data Presentation Modifications?**

Reviewer #1: all is clear

Reviewer #2: Accept

Reviewer #3: Minor revision

**Summary and General Comments**

Reviewer #1: The presented manuscript describes the results of a baseline cross-sectional study in Malawi to estimate the baseline STH prevalence and intensities as well as associated risk factors. 

The manuscript is very clearly written and presents solid data. 

The data again highlights the often underestimated importance of the adult population as a reservoir for STH and especially hookworm infections and further underpins the existing proof that links STH infections with education and wealth metrics. 

Another very interesting point that was briefly touched upon by the authors in the discussion was the fact they actually support the fact that (duplicate) Kato-Katz remains sufficiently sensitive to assess the distribution of STH and to link it with risk factors in this setting of very low STH infections (below WHO recommended thresholds for elimination of morbidity). In my opinion, this raises questions on the practical use/necessity of highly sensitive (but more complex and expensive) methods like qPCR to validate elimination of STH (in the sense of WHO or in the strict sense).

Reviewer #2: The study is a well designed and planned research. It is particularly significant in understanding the epidemiology and providing information that will aid decision making in the control of NTDs.

Reviewer #3: Separate file attached with details.

PLOS authors have the option to publish the peer review history of their article (what does this mean?). If published, this will include your full peer review and any attached files.

Reviewer #1: Yes: Johnny Vlaminck

Reviewer #2: Yes: Adebiyi Adeniran

Reviewer #3: No
---

## [Decision Letter · Decision Letter 1]

5 Mar 2021

Dear Mr Witek-McManus,

We are pleased to inform you that your manuscript 'Epidemiology of soil-transmitted helminths following sustained implementation of routine preventive chemotherapy: demographics and baseline results of a cluster randomised trial in southern Malawi' has been provisionally accepted for publication in PLOS Neglected Tropical Diseases.

Best regards,

Prof. María-Gloria Basáñez, PhD, MSc

Associate Editor

Aysegul Taylan Ozkan

Deputy Editor

Reviewer's Responses to Questions

**Key Review Criteria Required for Acceptance?**

**Methods**

-Are the objectives of the study clearly articulated with a clear testable hypothesis stated?

-Is the study design appropriate to address the stated objectives?

-Is the population clearly described and appropriate for the hypothesis being tested?

-Is the sample size sufficient to ensure adequate power to address the hypothesis being tested?

-Were correct statistical analysis used to support conclusions?

-Are there concerns about ethical or regulatory requirements being met?

Reviewer #2: Yes

Reviewer #3: (No Response)

**Results**

-Does the analysis presented match the analysis plan?

-Are the results clearly and completely presented?

-Are the figures (Tables, Images) of sufficient quality for clarity?

Reviewer #2: Yes

Reviewer #3: (No Response)

**Conclusions**

-Are the conclusions supported by the data presented?

-Are the limitations of analysis clearly described?

-Do the authors discuss how these data can be helpful to advance our understanding of the topic under study?

-Is public health relevance addressed?

Reviewer #2: Yes

Reviewer #3: (No Response)

**Editorial and Data Presentation Modifications?**

Reviewer #2: (No Response)

Reviewer #3: (No Response)

**Summary and General Comments**

Reviewer #2: (No Response)

Reviewer #3: The authors have addressed comprehensively all my earlier comments. I don't have further comments.

PLOS authors have the option to publish the peer review history of their article (what does this mean?). If published, this will include your full peer review and any attached files.

Reviewer #2: **Yes: **Adebiyi Adeniran

Reviewer #3: No

---

## [Editor Report · Acceptance letter]

6 May 2021

Dear Mr Witek-McManus,

We are delighted to inform you that your manuscript, "Epidemiology of soil-transmitted helminths following sustained implementation of routine preventive chemotherapy: demographics and baseline results of a cluster randomised trial in southern Malawi," has been formally accepted for publication in PLOS Neglected Tropical Diseases.

Best regards,

Shaden Kamhawi

co-Editor-in-Chief

Paul Brindley

co-Editor-in-Chief
